# Segmentation Control in Dynamic Wireless Charging for Electric Vehicles

**Tran Duc Hiep** [1,2], **Nguyen Huu Minh** [1], **Tran Trong Minh** [1], **Nguyen Thi Diep** [3] and **Nguyen Kien Trung** [1,*]

[1] School of Electrical and Electronic Engineering, Hanoi University of Science and Technology, Hanoi 11615, Vietnam; hieptd@haui.edu.vn (T.D.H.); minh.nh232211m@sis.hust.edu.vn (N.H.M.); minh.trantrong@hust.edu.vn (T.T.M.)
[2] School of Electrical and Electronic Engineering, Hanoi University of Industry, Hanoi 100000, Vietnam
[3] Faculty of Control and Automation, Electric Power University, Hanoi 100000, Vietnam; diepnt@epu.edu.vn
[*] Correspondence: trung.nguyenkien1@hust.edu.vn

**Abstract**

Dynamic wireless charging systems have emerged as a promising solution to extend the driving range of electric vehicles by enabling energy transfer while the vehicle is in motion. However, the segment-based charging lane structure introduces challenges such as pulsation of the output power and the need for precise switching control of the transmitting segments. This paper proposes a position-sensorless control method for managing transmitting lines in a dynamic wireless charging system. The proposed approach uses a segmented charging lane structure combined with two receiving coils and LCC compensation circuits on both the transmitting and receiving sides. Based on theoretical analysis, the study determines the optimal switching positions and signals to reduce the current fluctuation. To validate the proposed method, a dynamic wireless charging system prototype with a power rating of $3kW$ was designed, constructed, and tested in a laboratory environment. The results demonstrate that the proposed position-sensorless control method effectively mitigates power fluctuations and enhances the stability and efficiency of the wireless charging process.

**Keywords:** dynamic wireless charging; segmentation control; switching control



## 1. Introduction

Electric vehicles (EVs) are increasingly being recognized as a viable alternative to conventional gasoline-powered vehicles [1,2]. However, challenges such as prolonged charging durations and high initial costs remain significant barriers to widespread application [3,4]. A promising solution to these limitations is the implementation of dynamic wireless charging (DWC) systems, which enable energy transfer to moving vehicles, thereby reducing stationary charging demands and minimizing the need for large battery capacities. Studies suggest that equipping 20% of a roadway with a 40 kW charging infrastructure can extend an EV's driving range by at least 80% [5].

In a DWC system, inductive power transfer is achieved through transmitting coils embedded beneath the roadway and receiving coils installed on the vehicle. Transmitting coils are generally categorized into two types: long-track transmitting coils [6–9] and segmented transmitting coils [10–13]. The long-track configuration, when powered in areas without EV presence, leads to decreased energy transfer efficiency and increased electromagnetic field leakage. In contrast, the segmentation approach allows for selective activation, improving efficiency and mitigating stray electromagnetic emissions. Despite

these advantages, segmented coil systems introduce a higher system complexity and increased costs. Additionally, the switching of transmitting segments dynamically alters the mutual inductance between transmitting and receiving coils, causing power fluctuations that may induce irregular charging patterns and thermal stress, ultimately affecting battery life [14–16]. To mitigate these adverse effects, an optimized control strategy is required to regulate transmitting segment activation based on vehicle position. The objective is to minimize current fluctuation and enhance overall transmission efficiency, ensuring stable power delivery throughout the charging process.

To control the switching of transmitting coils in dynamic wireless charging systems, researchers have explored three main approaches: communication-based control, sensor-based control, and observer-based control. Each method has its advantages and limitations, particularly in terms of system complexity, reliability, and real-time performance.

- Communication-Based Control:

  This widely used method involves transmitting and receiving communications to manage charging segments, current, and voltage. This approach simplifies control but is primarily effective for low-speed vehicles and is prone to interference in strong magnetic field environments. In intelligent transportation systems, the characteristics of communication methods must be carefully considered to ensure reliability. Studies [17,18] propose a general framework for charging lane control; however, further research is required to refine its practical implementation. Additionally, studies [12,19] use charging power or current data to manage transmission segments via RF communication. While effective in theory, these studies fail to address key challenges such as communication delays and signal interference, which could compromise system performance in real-world applications.

- Sensor-Based Control:

  To improve precision, some researchers have proposed sensor-based solutions that determine a vehicle's location to control transmission segments. This approach offers greater accuracy but significantly increases system complexity and cost. For example, the method in [20,21] detects vehicle position using sensors and cameras, yielding reliable results but lacking a direct strategy for controlling transmitting lines. A more advanced approach in [22] employs sensor arrays along the charging lane to detect a vehicle's position and provide feedback for vehicle alignment; however, applying this system for segment control would require a more complex transmission-side structure. Another method, described in [13], introduces automatic switching via the QDDQ coil system, achieving 85% efficiency but suffering from the intricate design of transmitting coils. Alternatively, RF sensors are used in [23] to track EV speed and position, whereas refs. [13,24] integrate position sensors with specially designed coils to enable segment switching in DWC systems.

- Observer-Based Control:

  A third approach relies on observer-based techniques, which use the system's built-in hardware to estimate vehicle position. While this eliminates the need for additional sensors, it introduces complex mathematical models, making experimental implementation challenging. Furthermore, a common limitation across all three control methods is the lack of strategies to minimize power circuit switching when activating or deactivating transmitting modules. Studies [25–28] attempt to address this by using coupling coefficient variations between adjacent coils to determine vehicle position, but this method is best suited for systems with minimal coupling changes, such as electric trains. Other studies, such as [29], estimate vehicle position through power estimation and system transfer function modeling, while ref. [30] analyzes phase angles of input and reflected impedance. These techniques can support charging lane

control, but their high computational demands necessitate powerful controllers to ensure real-time performance.

It is evident that communication-based control and sensor-based hardware installations present significant drawbacks, including susceptibility to interference and increased system complexity, compared to observer-based methods. To address these challenges, this paper proposes an approach for determining the activation and deactivation positions of transmitting segments and their corresponding switching signals. This method relies solely on current measurements from the transmitting coils, eliminating the need for additional complex hardware. The proposed methodology was validated using a 3 kW dynamic wireless charging system model in a laboratory setting, where the charging current fluctuation was 2.2% in simulation and 3.3% in the experimental results, demonstrating the effectiveness of the proposed approach. This study is organized as follows: The first section offers an overview of dynamic wireless charging systems, highlighting the key challenges in controlling transmission lines and reviewing current research trends. The second section details the structure of the dynamic charging system, presents the mathematical foundations supporting the proposed control method, and outlines the switching strategy. The third section presents both simulation and experimental results, then demonstrates the feasibility and performance of the proposed algorithm. The final section discusses the results achieved, addresses the limitations of the study, and proposes future research.

## 2. System Structure and Theoretical Analysis

### 2.1. Proposed System Structure

The proposed DWC system structure for EVs is shown in Figure 1. There are two parts of the DWC system: the transmitting side and the receiving side. The transmitting side consists of coils with identical specifications placed next to each other to form a charging lane. An inverter unit is used to supply power to three transmitting coils through primary compensation circuits, forming modules that help reduce the number of power converters. Each module operates with a particular dedicated controller, and these controllers are interconnected through a communication network. The receiving side comprises two coils spaced 1.5 times the length of a transmitting coil, which helps minimize current fluctuations and enhance power transfer [15]. These coils are connected to secondary compensation circuits and a rectifier, enabling efficient charging of the vehicle's energy storage system. To achieve a stable output, LCC compensation circuits are employed on both transmitting and receiving sides to maintain a constant current delivered to the EV battery. The detailed coil structure of the proposed system is presented in Figure 2. Figure 2a illustrates the 3D structure of the coils, where each coil is designed with three layers: the first layer consists of rectangular Litz wire coils, the second layer comprises ferrite plates to direct magnetic flux and increase the coupling coefficient, and the third layer includes aluminum plates that serve to prevent the magnetic field from dispersing into the environment. Figure 2b shows the 2D arrangement of the coils along the charging lane, where the transmitting coils are sequentially numbered. The receiver coil position is referenced along the $x$-axis, with $x = 0$ defined when the center of receiver coil R1 aligns with the center of transmitter coil 1. The main coil specifications are summarized in Table 1.

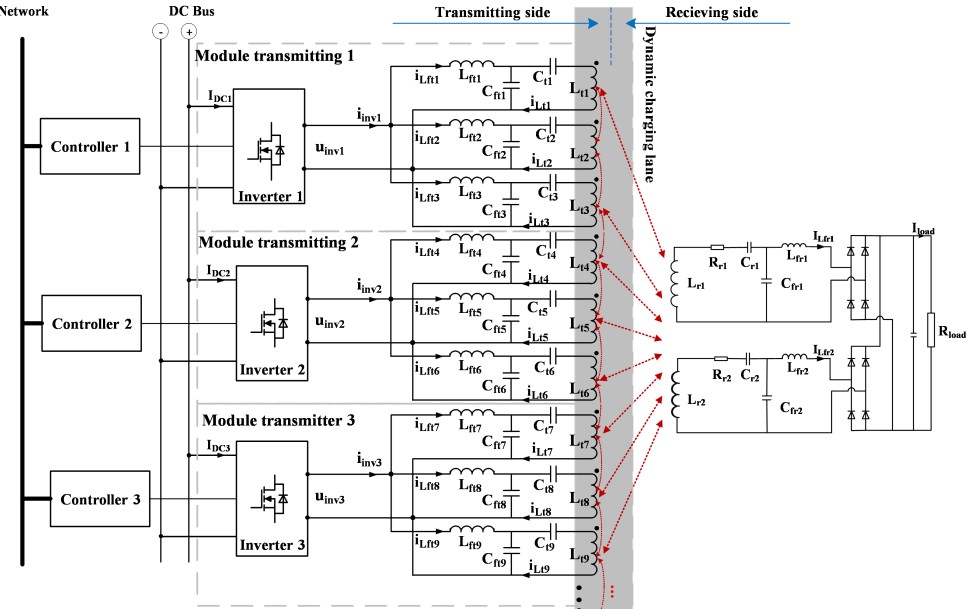

**Figure 1.** Structure of the proposed dynamic wireless charging system.

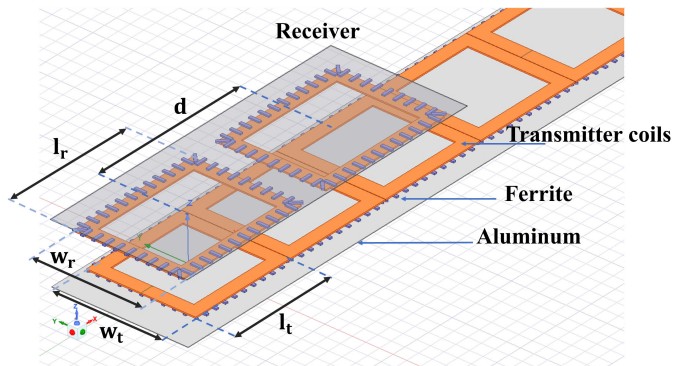

(**a**) Three-dimensional structure of the coils

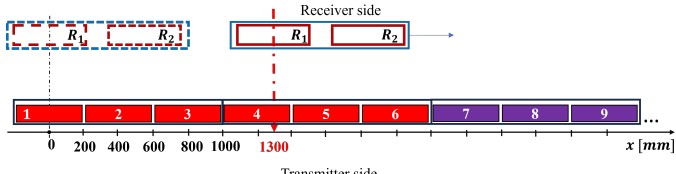

(**b**) Two-dimensional structure of the coils on the charging lane

**Figure 2.** Structure of the coils in the proposed DWC system.

**Table 1.** Magnetic coupler parameters.

| Symbol | Parameter | Definition |
|---|---|---|
| $L_{t_1} = L_{t_2} = ... = L_{t_i} = L_t$ | 102 µH | Transmitting coil inductance |
| $R_{t_1} = R_{t_2} = ... = R_{t_i} = R_t$ | 0.27 Ω | Transmitting coil resistance |
| $L_{r_1} = L_{r_2} = L_r$ | 120 µH | Receiving coil inductance |
| $R_{r_1} = R_{r_2} = R_r$ | 0.37 Ω | Receiving coil resistance |
| $l_t$ | 400 mm | Transmitting coil length |
| $w_t$ | 400 mm | Transmitting coil width |
| $l_r$ | 500 mm | Receiving coil length |
| $w_r$ | 400 mm | Receiving coil width |
| $d$ | 600 mm | Distance between receiving coils |
| $h$ | 150 mm | Air gap |

The DWC system needs to use a compensation circuit to create resonance on the transmitting and receiving sides to improve transmission efficiency. The capacitors and compensation inductors are designed to generate a constant current on the load. The parameters of the compensation circuit are shown in Table 2.

**Table 2.** Compensation circuits parameters.

| Parameter | Value | Parameter | Value |
|-----------|-------|-----------|-------|
| $L_{ft_i}$ | 100 µH | $L_{fr_i}$ | 11.4 µH |
| $C_{ft_i}$ | 0.035 µF | $C_{fr_i}$ | 0.307 µF |
| $C_{t_i}$ | 0.025 µF | $C_{r_i}$ | 0.0323 µF |

*2.2. Coupling Coefficient Analysis*

In the designed dynamic wireless charging system, under the condition that the vehicle moves along a straight lane with a constant distance between the vehicle and the road surface, the charging current depends on two factors: the operating frequency of the system and the mutual inductance between the transmitting coils embedded in the road and the receiving coils mounted on the vehicle. As the vehicle moves over modular charging segments with identical design, the mutual inductance exhibits a repetitive pattern, referred to as the mutual inductance cycle [15]. In wireless power transfer systems such as the one considered in this study, the mutual inductance is commonly expressed in terms of the coupling coefficient $k$, as shown in (1).

$$k = \frac{M}{\sqrt{L_t L_r}} \tag{1}$$

where $L_t$ and $L_r$ are the self-inductances of the transmitting and receiving coils, respectively, and $M$ is the mutual inductance between them.

Similarly, the coupling coefficients between the transmitting coils and receiving coil 1 ($r_1$), receiving coil 2 ($r_2$), and between adjacent transmitting coils ($t_i$), based on the self- and mutual inductances, are denoted as $k_{t_i r_1}(x)$, $k_{t_i r_2}(x)$, and $k_{t_i t_{i+1}}(x)$, respectively, and expressed as follows:

$$k_{t_i r_1}(x) = \frac{M_{t_i r_1}(x)}{\sqrt{L_{t_i} L_{r_1}}} = \frac{M_{t_i r_1}(x)}{\sqrt{L_t L_r}} \tag{2}$$

$$k_{t_i r_2}(x) = \frac{M_{t_i r_2}(x)}{\sqrt{L_{t_i} L_{r_2}}} = \frac{M_{t_i r_2}(x)}{\sqrt{L_t L_r}} \tag{3}$$

$$k_{t_i t_{i+1}}(x) = \frac{M_{t_i t_{i+1}}(x)}{\sqrt{L_{t_i} L_{t_{i+1}}}} = \frac{M_{t_i t_{i+1}}(x)}{L_t} \tag{4}$$

Here, $M_{t_i r_1}(x)$, $M_{t_i r_2}(x)$, and $M_{t_i t_{i+1}}(x)$ denote the mutual inductances between the $i^{th}$ transmitting coil and receiving coil 1, receiving coil 2, and the adjacent transmitting coil ($i + 1$), respectively. The index i refers to the transmitting coil number, with $i = 1$ to $n$; $L_t$ and $L_r$ are defined as shown in Table 1; $x$ is the position of the receiving coil, as defined in Figure 2b.

The total coupling coefficient of the transmitting coils with receiving coils $r_1$ and $r_2$ is defined as in the following equations:

$$k_{tr_1}(x) = \sum_{i=1}^{n} k_{t_i r_1}(x) \tag{5}$$

$$k_{tr_2}(x) = \sum_{i=1}^{n} k_{t_i r_2}(x) \tag{6}$$

Since the operating frequency of the system is significantly higher than the fluctuation frequency of the coupling coefficient, the switching process of power electronic converters in the system can be considered in a steady-state condition during dynamic charging. Therefore, the fluctuation of the charging current mainly depends on the coupling coefficient between the transmitting and receiving coils in the dynamic wireless charging system. Based on the observation, this paper proposes an analysis of the variation rule of the coupling coefficient function to control the switching of power transfer segments.

Modeling the DWC system using the circuit analysis methods is highly complex due to numerous uncertainties such as vehicle speed, lateral deviation, and the influence of the battery system. To simplify the analysis, this study employed the finite element simulation method to determine the coupling coefficient function. Ansys Maxwell software 2021R1 was used to simulate the coil system, where the coil model adopted the structure described in Figure 2. Due to the spatial limitations of the simulation environment, the number of transmitting coils considered in this case is 9 coils ($i$ = 1 to 9). The simulation yields the coupling coefficient values as a function of vehicle position ($x$), as shown in Figure 3.

Specifically, Figure 3a,b present the simulated coupling coefficients between the transmitting coils ($t_1$ to $t_9$) and receiving coils $r_1$ and $r_2$, while Figure 3c shows the coupling coefficients between adjacent transmitting coils. The results indicate that the coupling coefficient curves between each transmitting coil and the receiving coils ($r_1$ or $r_2$) follow the same pattern. Due to the movement of the receiver coils, the coupling coefficient (or mutual inductance) between the transmitter and receiver coils can be either positive or negative, depending on the relative position and orientation of the coils. Specifically, the coupling coefficient of the $i$-th transmitting coil with the receiving coil reaches its maximum value when the center of that transmitting coil is perfectly aligned with the center of the receiving coil. As the receiving coil moves away from this alignment, the coupling coefficient gradually decreases. Moreover, the peaks of the coupling coefficient are spaced at 400 mm, corresponding to the center-to-center distance between adjacent transmitting coils. The average total coupling coefficient between the transmitting coils and each receiving coil ($k_{tr1}, k_{tr2}$) is around 0.14. In addition, the coupling between adjacent transmitting coils is also considerable and included in the system analysis.

For analytical clarity, the coupling coefficients simulated in Figure 3 are approximated by the functions given in (7)–(9). These functions, which describe the coupling coefficient as a function of the electric vehicle's position $x$, are identified using the System Identification Toolbox in MATLAB and Simulink R2024a. The corresponding coefficients are summarized in Table 3.

$$k_{t_i r_1}(x) = r \cdot \left[ a_1 \cdot e^{\frac{-(x+(5-i)\Delta x_t - b_1)^2}{c_1^2}} + a_2 \cdot e^{\frac{-(x+(5-i)\Delta x_t - b_2)^2}{c_2^2}} \right] \tag{7}$$

$$k_{t_i r_2}(x) = r \cdot \left[ a_1 \cdot e^{\frac{-(x+\Delta x_r+(5-i)\Delta x_t - b_1)^2}{c_1^2}} + a_2 \cdot e^{\frac{-(x+\Delta x_r+(5-i)\Delta x_t - b_2)^2}{c_2^2}} \right] \tag{8}$$

$$k_{t_i t_{i+1}}(x) = -c - b \cdot \left[ a_1 \cdot e^{\frac{-(x-\Delta x_0+(5-i)\Delta x_t - b_1)^2}{c_1^2}} + a_2 \cdot e^{\frac{-(x-\Delta x_0+(5-i)\Delta x_t - b_2)^2}{c_2^2}} \right] \tag{9}$$

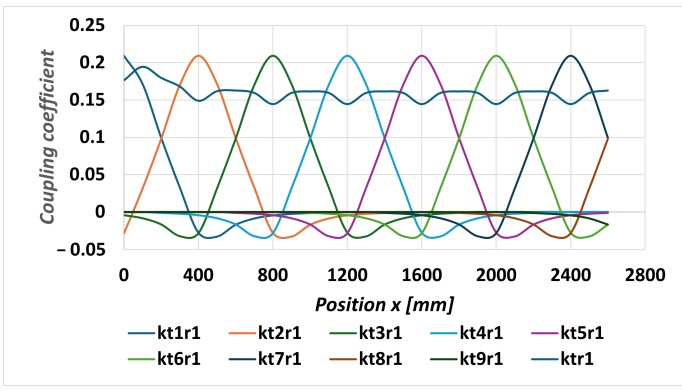

(**a**) The coupling coefficient of charging lane to receiving coil $r_1$

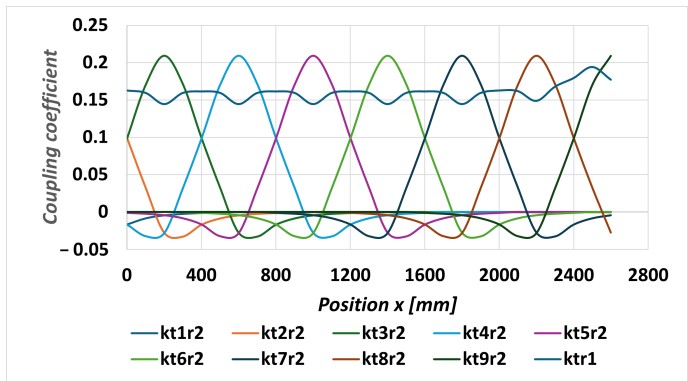

(**b**) The coupling coefficient of charging lane to receiving coil $r_2$

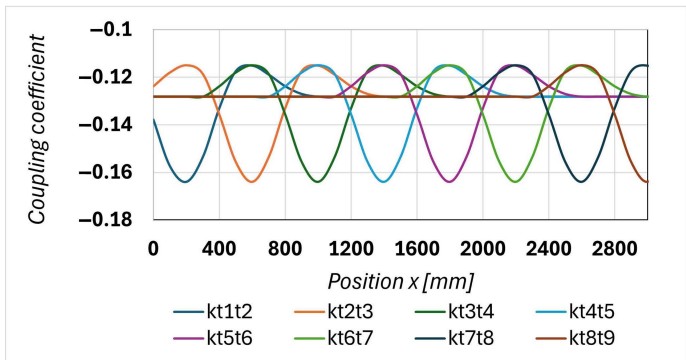

(**c**) The coupling coefficient of adjacent transmitting coils

**Figure 3.** The coupling coefficient of the transmitting side to the receiving side.

**Table 3.** Coefficients of the coupling coefficient approximation functions.

| Parameter | Value | Parameter | Value |
|---|---|---|---|
| $r$ | 0.7452155756 | $b$ | 0.1202546279 |
| $a_1$ | 0.175 | $c$ | 0.12 |
| $a_2$ | 0.1602 | $\Delta x_0$ | 200 |
| $b_1$ | 1522 | $\Delta x_t$ | 400 |
| $b_2$ | 1705 | $\Delta x_r$ | 600 |
| $c_1$ | 145.6 | $c_2$ | 139.8 |

Since the two receiving coils are positioned relatively far apart, with a center-to-center distance of 600 mm (from the center of receiving coil $r_1$ to the center of receiving coil $r_2$), the mutual coupling coefficient between them is approximately zero and, thus, neglected in this analysis.

### 2.3. Proposed Switching Method

In the proposed system, the transmitting side of the charging lane is designed in a modular structure to enhance power transfer efficiency and reduce magnetic field radiation. Consequently, it is necessary to control the activation and deactivation of the transmitting modules based on the position of the electric vehicle. This section presents a switching control strategy between adjacent transmitting segments to minimize load power pulsation.

The equivalent circuit of Figure 1 is presented in Figure 4. The electromagnetic relationships between the coils are represented by the induced voltages, where the induced voltage in each coil depends on the mutual inductance and the current in the other coils. Each transmitting coil is electromagnetically coupled to the other transmitting coils as well as to the two receiving coils. The receiving coils are electromagnetically coupled only to the transmitting coils. Since the receiving coils are positioned relatively far apart, the induced voltage between them is negligible and, therefore, ignored in the analysis.

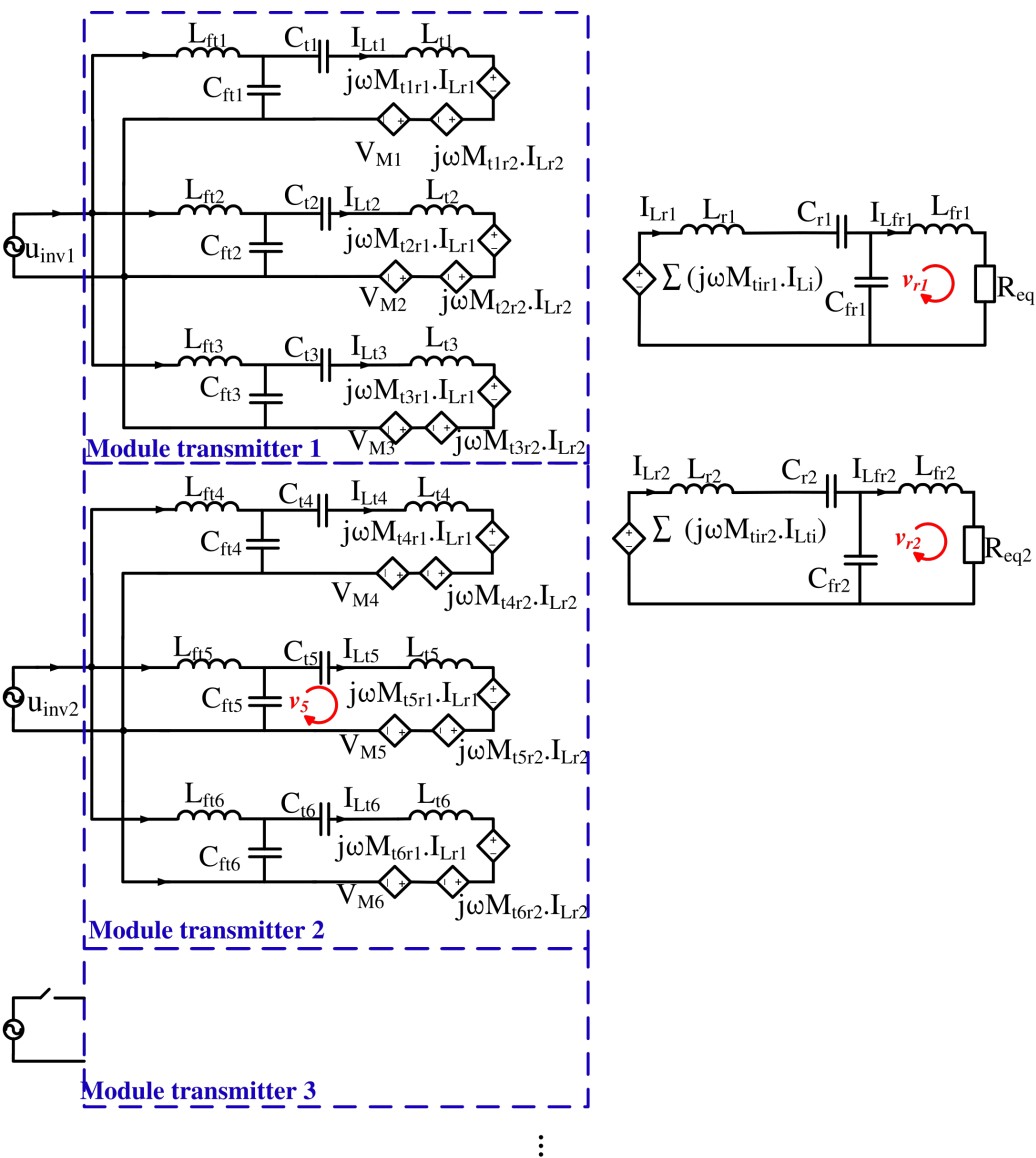

**Figure 4.** Equivalent circuit.

Here, the transmitting coils are identically designed, and the two receiving coils are also identical. Assuming the transmission lane is infinitely long, the compensation circuits on the transmitting side are designed identically, and the compensation circuits on the

receiving side are also identical. Therefore, the compensation circuits are designed to satisfy the following resonance condition:

$$
\begin{cases}
j\omega L_{fti} + \dfrac{1}{j\omega C_{fti}} = 0 \\[2mm]
j\omega L_{fr} + \dfrac{1}{j\omega C_{fr}} = 0 \\[2mm]
\dfrac{1}{j\omega C_{fr}} + \dfrac{1}{j\omega C_r} + j\omega L_r = 0 \\[2mm]
\dfrac{1}{j\omega C_{fti}} + j\omega L_{ti} + \dfrac{1}{j\omega C_{ti}} + V_{Mi} = 0
\end{cases}
\tag{10}
$$

Here,

$$
\begin{cases}
L_{t_1} = L_{t_2} = ... = L_{t_i} = ... = L_t \\
C_{ft1} = C_{ft2} = ... = C_{fti}... = C_{ft} \\
C_{t1} = C_{t2} = ... = C_{ti}... = C_t \\
L_{r_1} = L_{r_2} = L_r \\
C_{fr1} = C_{fr2} = C_{fr} \\
C_{r1} = C_{r2} = C_r
\end{cases}
\tag{11}
$$

The currents in the transmitting coils are determined as follows [11]:

$$
I_{L_{t_1}} = I_{L_{t_2}} = ... = I_{L_{t_i}} = ... = I_{L_t} = -j\omega C_{ft} \cdot U_{inv} = j\frac{U_{inv}}{\omega L_{ft}}
\tag{12}
$$

By analyzing the loops $v_{r1}$ and $v_{r2}$ in Figure 4 using Kirchhoff's Voltage Law, the following expressions are obtained:

$$
\begin{cases}
I_{L_{fr1}} = j\frac{\omega L_{fr1}}{R_{eq1}} I_{L_{r1}} = j\frac{\omega L_{fr}}{R_{eq}} I_{L_{r_1}} \\[2mm]
I_{L_{fr2}} = j\frac{\omega L_{fr2}}{R_{eq2}} I_{L_{r2}} = j\frac{\omega L_{fr}}{R_{eq}} I_{L_{r_2}}
\end{cases}
\tag{13}
$$

$$
\begin{cases}
I_{L_{r_1}} = \dfrac{R_{eq_1}}{(\omega L_{fr1})^2} \cdot \displaystyle\sum_{i=1}^{n} \left[j\omega M_{t_i r_1} I_{L_{t_i}}\right] = j\frac{R_{eq}\sqrt{L_t L_r}}{\omega L_{fr}^2} \cdot \displaystyle\sum_{i=1}^{n} k_{t_i r_1} \cdot I_{L_t} \\[4mm]
I_{L_{r_2}} = \dfrac{R_{eq_2}}{(\omega L_{fr2})^2} \cdot \displaystyle\sum_{i=1}^{n} \left[j\omega M_{t_i r_2} I_{L_{t_i}}\right] = j\frac{R_{eq}\sqrt{L_t L_r}}{\omega L_{fr}^2} \cdot \displaystyle\sum_{i=1}^{n} k_{t_i r_2} \cdot I_{L_t}
\end{cases}
\tag{14}
$$

where $R_{eq_1}, R_{eq_2}$ denote the equivalent AC loads:

$$
R_{eq_1} = R_{eq_2} = R_{eq} = \frac{16}{\pi^2} R_{load}
\tag{15}
$$

The load current is determined as follows:

$$
I_{load} = \frac{\pi}{2\sqrt{2}}(I_{L_{fr_1}} + I_{L_{fr_2}})
\tag{16}
$$

From Equations (12) to (16), the load current can be derived as given in Equation (17) below:

$$
I_{load} = \frac{\pi}{2\sqrt{2}} \frac{U_{inv}}{\omega L_{ft} L_{fr}} \sqrt{L_r L_t} \cdot \left[\sum_{i=1}^{n} k_{t_i r_1} + \sum_{i=1}^{n} k_{t_i r_2}\right]
\tag{17}
$$

From Equation (17), it can be observed that when the system parameters ($U_{inv}$, $\omega$, $L_{ft}$, $L_{fr}$, $L_r$, $L_t$) are fixed, the load current depends solely on ($\sum_{i=1}^{n} k_{t_i r_1} + \sum_{i=1}^{n} k_{t_i r_2}$). This term, in turn, depends on the position of the electric vehicle ($x$), where coils in the transmitting lane are switched on or off. The objective of this study is to determine the optimal switching instants of these modules to minimize the ripple in the load current.

In this system, the dynamic wireless charging is designed in a modular structure, so instead of turning on and off each coil, it is turned on and off by each module. Moreover, considering the dimensions of the transmitting and receiving coils as described in Table 1 and as shown in Figure 2, it is necessary to simultaneously activate at least two consecutive transmitting modules. This ensures proper alignment with the receiving coil size and accommodates the movement speed of the secondary side. In addition, an efficient switching strategy is required to minimize current transients and reduce load current ripple. Consider the first three modules of the charging lane, as illustrated in Figures 1 and 2. To ensure continuous power delivery to the electric vehicle, modules 1 and 2 are activated as soon as the vehicle enters the charging lane. When the vehicle reaches the switching position, module 1 is deactivated and module 3 is activated. Based on the load current in (17), the pulsation of the load current during the switching transition from module 1 to module 3 is shown below:

$$\Delta I_{load}(x) = I_{load_{module2,3}}(x) - I_{load_{module1,2}}(x) = \frac{\pi}{2\sqrt{2}} \frac{U_{inv}}{\omega L_{ft} L_{fr}} \sqrt{L_r L_t} \cdot \Delta k(x) \tag{18}$$

Here,

$$\Delta k(x) = \sum_{i=4}^{9} k_{t_i r_1}(x) + \sum_{i=4}^{9} k_{t_i r_2}(x) - \sum_{i=1}^{6} k_{t_i r_1}(x) - \sum_{i=1}^{6} k_{t_i r_2}(x)$$
$$= k_{t_7 r_1}(x) + k_{t_8 r_1}(x) + k_{t_9 r_1}(x) + k_{t_7 r_2}(x) + k_{t_8 r_2}(x) + k_{t_9 r_2}(x)$$
$$- k_{t_1 r_1}(x) - k_{t_2 r_1}(x) - k_{t_3 r_1}(x) - k_{t_1 r_2}(x) - k_{t_2 r_2}(x) - k_{t_3 r_2}(x) \tag{19}$$

Based on Equations (7) and (8), the variation of $\Delta k$ with respect to the vehicle position $x$, as expressed in Equation (19), is illustrated in Figure 5. The results show that $\Delta k$ is approximately zero when the vehicle position $x$ is within the range of 1200 mm to 1400 mm. This indicates that, within this range, the load current ripple is almost eliminated during the switching transition from transmitting module 1 to transmitting module 3.

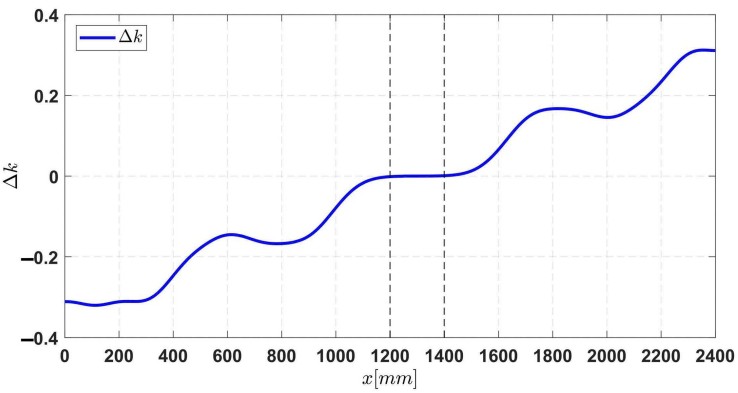

**Figure 5.** Characteristics of $\Delta k$ according to the position of EV ($x$).

The current challenge is to determine when the electric vehicle reaches this position without using any position-detection sensors. Considering the case where transmitting modules 1 and 2 are activated, and based on the equivalent circuit diagram shown in Figure 4, Kirchhoff's law is applied to loop $v_5$:

$$\left(\frac{1}{j\omega C_{t_5}} + j\omega L_{t_5}\right) I_{L_{t_5}} + j\omega M_{t_5 r_1}(x) I_{Lr_1} + j\omega M_{t_5 r_2}(x) I_{Lr_2} + j\omega M_{t_6 t_5}(x) I_{L_{t_6}}$$
$$+ j\omega M_{t_4 t_5}(x) I_{L_{t_4}} - \frac{1}{j\omega C_{ft_5}} \left(I_{Lft_5} - I_{L_{t_5}}\right) = 0 \tag{20}$$

From Equations (2) to (4), (12), (14), and (20), the current equation for compensation inductor $L_{ft5}$ can be derived as follows:

$$I_{L_{ft_5}}(x) = A\left[B\left(k_{t_5r_1}(x)\sum_{i=1}^{6}k_{t_ir_1}(x) + k_{t_5r_2}(x)\sum_{i=1}^{6}k_{t_ir_2}(x)\right) + C\left(k_{t_5t_6}(x) + k_{t_4t_5}(x)\right)\right] \quad (21)$$

By a similar analysis, the current equation for compensation coils $L_{ft4}$ and $L_{ft6}$ is derived as follows:

$$I_{L_{ft_4}}(x) = A\left[B\left(k_{t_4r_1}(x)\sum_{i=1}^{6}k_{t_ir_1}(x) + k_{t_4r_2}(x)\sum_{i=1}^{6}k_{t_ir_2}(x)\right) + C\left(k_{t_3t_4}(x) + k_{t_4t_5}(x)\right)\right] \quad (22)$$

$$I_{L_{ft_6}}(x) = A\left[B\left(k_{t_6r_1}(x)\sum_{i=1}^{6}k_{t_ir_1}(x) + k_{t_6r_2}(x)\sum_{i=1}^{6}k_{t_ir_2}(x)\right) + Ck_{t_5t_6}(x)\right] \quad (23)$$

where

$$A = -j\omega^3 C_{ft_i}^2 U_{inv}; \quad B = j\frac{R_{eq}L_rL_t}{\omega L_{fr}^2}; \quad C = L_t$$

From (21)–(23), it can be seen that the current on the compensating coils depends on the value of the mutual inductance. The value of these mutual inductances depends on the EVs' position.

Based on Equations (7)–(9), and (21)–(23), the current characteristics of the compensation inductors $I_{L_{ft_4}}$, $I_{L_{ft_5}}$ and $I_{L_{ft_6}}$ can be derived as Figure 6. The analysis results indicate that the position $x$ within the range of 1200 mm to 1400 mm can be accurately determined by measuring and comparing the currents in these inductors. Specifically, the position $x = 1200$ mm is identified when $I_{L_{ft_6}}$ exceeds $I_{L_{ft_5}}$ and simultaneously $I_{L_{ft_4}}$ reaches its maximum value. This condition is introduced to avoid confusion with the position $x = 1800$ mm. Similarly, the position $x = 1400$ mm is determined when $I_{L_{ft_5}}$ exceeds $I_{L_{ft_4}}$ and $I_{L_{ft_6}}$ reaches its maximum value, thus distinguishing it from $x = 800$ mm. For $x = 1300$ mm, it is sufficient to compare $I_{L_{ft_6}}$ and $I_{L_{ft_4}}$; when $I_{L_{ft_6}}$ is greater than $I_{L_{ft_4}}$, $x = 1300$ mm is identified without requiring additional conditions. To simplify and enhance control performance, the position $x = 1300$ mm is selected as the switching point from module 1 to module 3.

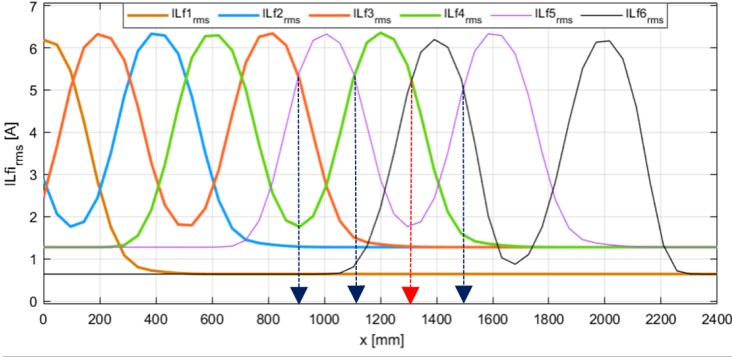

**Figure 6.** The characteristics show the calculated RMS value of the current in the compensation inductor as a function of the EV position.

## 3. Simulation and Experimental Results

### 3.1. Simulation Results

To validate the proposed method, a 3 kW simulation model was developed using MATLAB/Simulink. The system structure follows the configuration shown in Figure 1. The system parameters and compensation circuit parameters are provided in Tables 1 and 2. The load is modeled by a resistive load and the simulation process of the vehicle moving on a straight lane at 76.8 Km/h.

To evaluate the load current overshoot during the switching process from module 1 to module 3 at $x = 1300$ mm, a module switching algorithm is implemented as follows:

- Step 1: Measure the RMS current on the compensation inductors $L_{ft4}$ and $L_{ft6}$.
- Step 2: Compare the RMS current values on both coils. If $I_{L_{ft_6}} > I_{L_{ft_4}}$, module 1 is deactivated, and module 3 is activated.

The simulation results of the current according to the position of the transmission coil when performing the switching are shown in Figure 7a. The results show that the load current fluctuation is 2.2%. To provide a comprehensive basis for evaluating the switching at position $x = 1300$ mm, switching at other positions was also implemented, and the corresponding results were recorded. The simulation result for switching $x = 1500$ mm is shown in Figure 7b. In this case, the switching point is determined by comparing $I_{L_{ft_5}}$ and $I_{L_{ft_6}}$; switching is performed when $I_{L_{ft_5}} > I_{L_{ft_6}}$. The current pulsation in this case is 5.3%. Meanwhile, Figure 7c shows the outcome at 900 mm, where switching is triggered when $I_{L_{ft_5}} > I_{L_{ft_3}}$; the measured current pulsation in this case is 19.8%. These results demonstrate that switching at $x = 1300$ mm provides the most favorable current characteristics with the lowest pulsation.

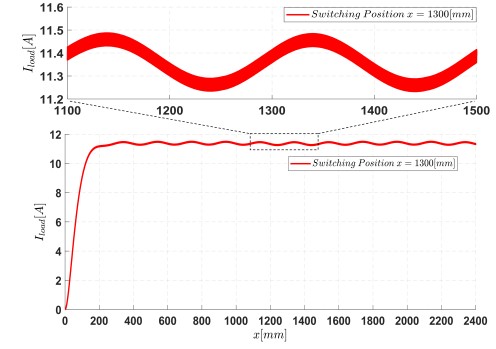

(**a**) Switching position at 1300 mm

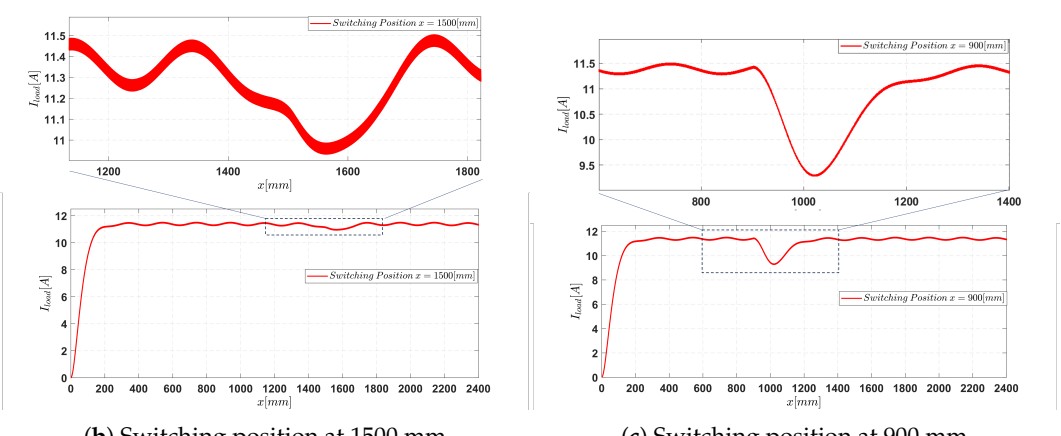

(**b**) Switching position at 1500 mm      (**c**) Switching position at 900 mm

**Figure 7.** Output current form at switching positions from simulation.

The simulation results of the load power characteristics for the aforementioned cases are presented in Figure 8. The results indicate that switching at the position $x = 1300$ mm yields the lowest power ripple, at only 5.8%. At neighboring positions such as $x = 1500$ mm, the power ripple increases to 9%. Notably, switching at $x = 900$ mm results in the highest power ripple, approximately 69.3%. Based on these findings, it can be concluded that $x = 1300$ mm is the most suitable switching position, as it significantly minimizes the power ripple during the energy transfer process.

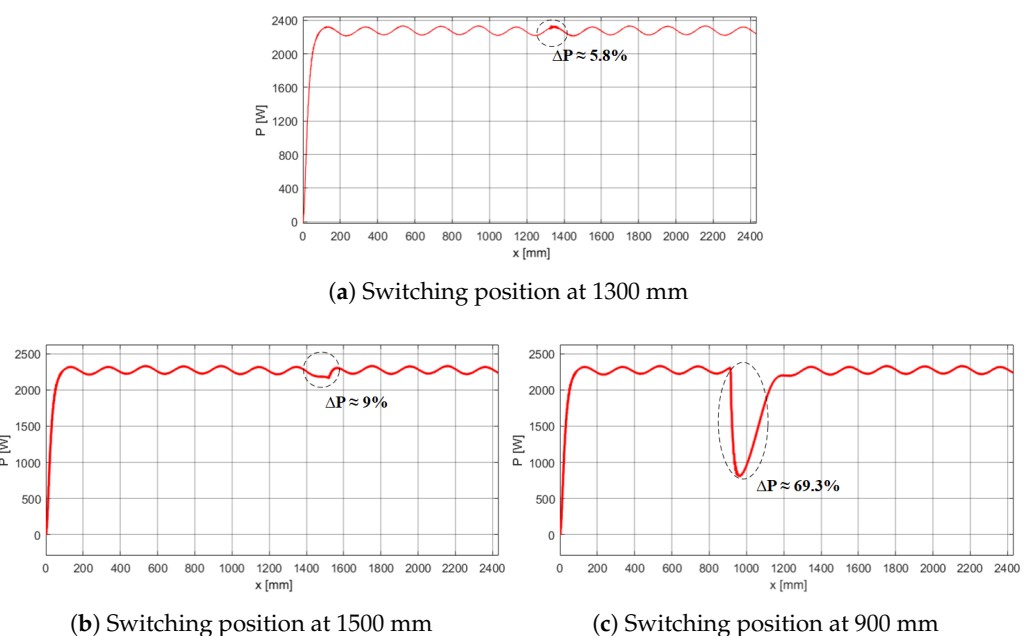

(**a**) Switching position at 1300 mm

(**b**) Switching position at 1500 mm

(**c**) Switching position at 900 mm

**Figure 8.** Output power characteristic curve at different switching positions.

### 3.2. Experimental Results

A 3 kW dynamic wireless charging system model for electric vehicles was developed in the laboratory, as shown in Figure 9, with system parameters listed in Tables 1 and 2. The system consists of two main parts: the transmitting side and the receiving side. The transmitting side includes a charging lane composed of nine transmitting coils, arranged into three modules. Each module consists of three transmitting coils connected to three LCC compensation circuits and shares a single inverter. The receiving side consists of two phases, each comprising one receiving coil, one LCC compensation circuit, and one rectifier. The two phases are connected in parallel and linked to the load. These phases are mounted on a vehicle model, which is placed on a linear motion structure that allows the vehicle to move along the charging lane. Additionally, the receiving coils can be adjusted laterally to simulate vehicle misalignment from the lane center, thereby emulating dynamic charging conditions as described in the theoretical analysis. Owing to spatial constraints within the laboratory environment, the experiments in this study were conducted at a limited speed of approximately 20 km/h. Based on the theoretical analysis, the switching position and the switching position indication based on the inductor current are independent of the movement speed.

The experimental gate–source (GS) pulse waveforms of two MOSFETs in the same inverter branch are shown in Figure 10. The output voltage and current waveforms of a single inverter module are presented in Figure 11, while the voltage waveforms across the transmission coils are illustrated in Figure 12. The results confirm that the inverter operates at the designed frequency of 85 kHz, and soft-switching is successfully achieved with the specified circuit parameters.

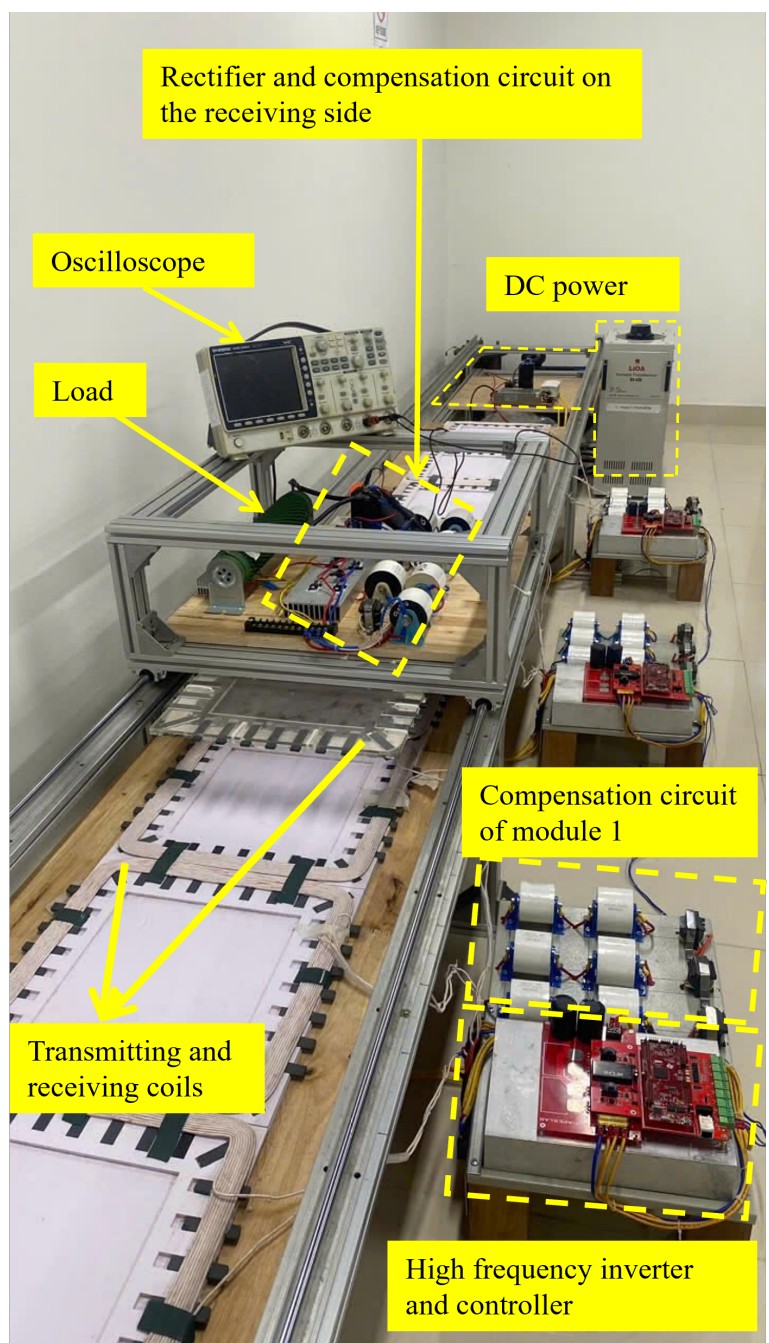

**Figure 9.** The experimental model of the dynamic wireless charging system for EVs.

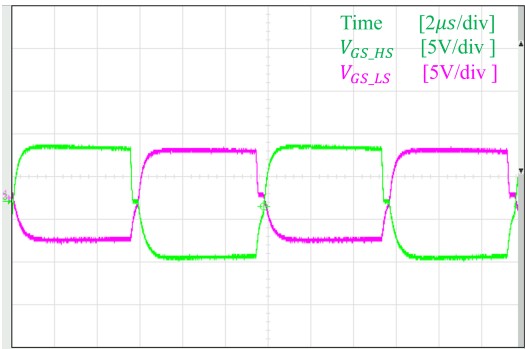

**Figure 10.** The GS pulse waveform on two switches of the same branch.

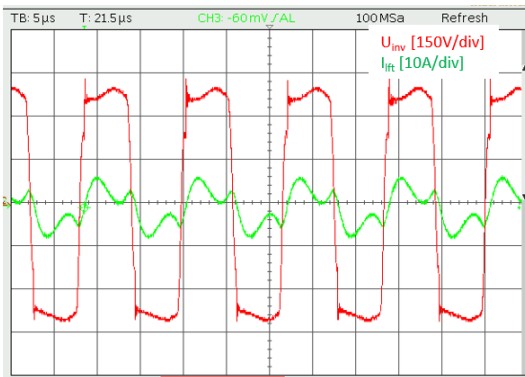

**Figure 11.** The output voltage and current of the inverter.

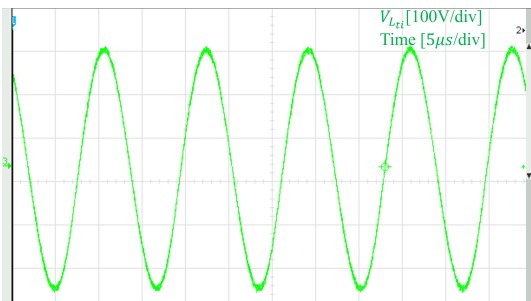

**Figure 12.** The voltage waveform on the transmitting coil.

To evaluate the constant current charging capability of the proposed design, a test scenario is constructed as follows: while the vehicle remains on the charging lane, the load resistance is varied to 5.6 Ω, 8.7 Ω, and 16 Ω, respectively. The corresponding voltage and current waveforms at the load are measured and shown in Figure 13. The experimental results of the measured voltage and current waveforms at the load indicate that as the load impedance increases, the load voltage rises accordingly. Meanwhile, the charging current remains stable with a deviation of only about 2%. These findings confirm that the proposed system achieves constant current charging capability, ensuring reliable power transfer even under varying load conditions.

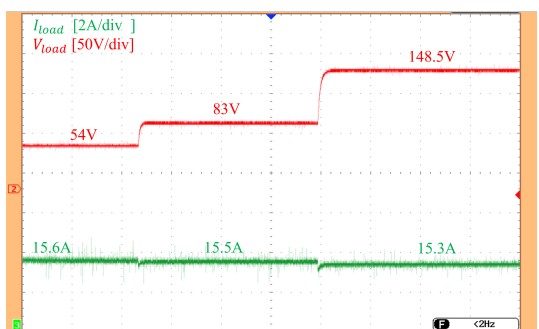

**Figure 13.** The Load current and voltage waveforms under varying load conditions.

To evaluate the capability of the proposed design to reduce charging current fluctuation during switching at optimal position, a test scenario is constructed as follows: the vehicle model is moved along the charging lane, and switching is performed at 900 mm, 1300 mm, and 1500 mm, as determined from the theoretical analysis. The waveforms of the load current (green), the current in the transmitter coil $L_{t_2}$ of module 1 (red), and the current in the transmitter coil $L_{t_7}$ of module 2 (purple) are shown in Figure 14.

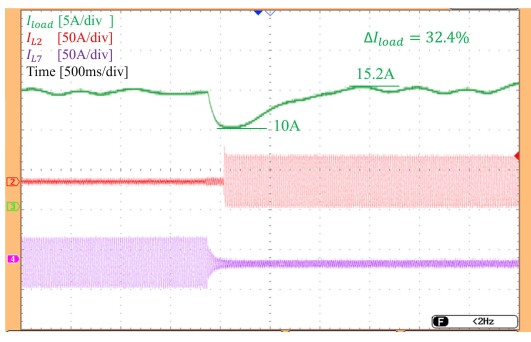

(**a**) Switching position at 900 mm

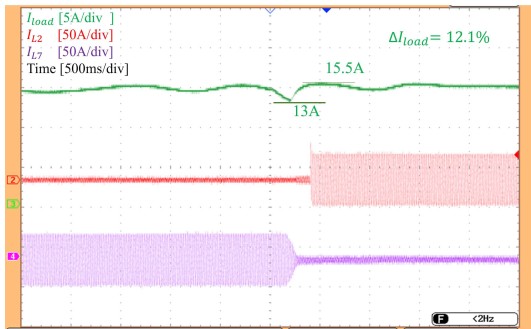

(**b**) Switching position at 1500 mm

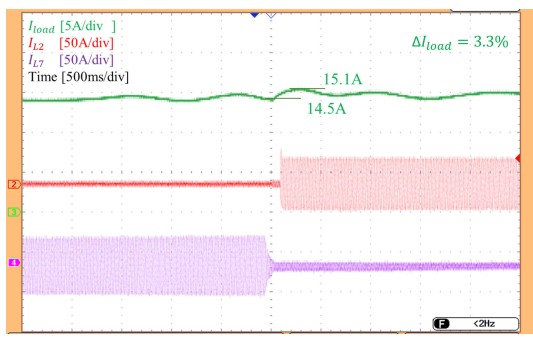

(**c**) Switching position at 1300 mm

**Figure 14.** Output current form at switching positions.

The experimental results show that current fluctuation during switching is relatively high when switching at $x \approx 900$ mm with $I_{L_{f_5},rms} > I_{L_{f_3},rms}$, reaching approximately 32.4%. When switching at $x \approx 1500$ mm with $I_{L_{f_5},rms} > I_{L_{f_6},rms}$, the output current fluctuation is about 12.1%. In the case of switching at $x \approx 1300$ mm, where $I_{L_{f_6},rms} > I_{L_{f_4},rms}$, which is the proposed solution, the output current fluctuation is minimized to approximately 3.3%. Despite minor deviations due to measurement and system uncertainties, the results confirm that switching at 1300 mm yields the lowest current fluctuation, consistent with theoretical analysis.

To evaluate the adaptability of the proposed design, a test scenario was constructed as follows: the vehicle model was set to move in a straight line while the receiver coil was shifted laterally toward the edge of the lane to simulate off-center movement. The current waveforms for both cases—vehicle moving along the centerline and with a 10cm lateral offset (equivalent to 50% misalignment)—are shown in Figure 15. The results indicate that the current waveform in the off-center case is similar to that of the centered straight movement; however, the charging current is significantly reduced from approximately 15 A to 7.8 A. This reduction can be attributed to the significant drop in coupling coefficient caused by the misalignment between transmitting and receiving coils, as theoretically analyzed.

These results demonstrate that the proposed switching method remains applicable under off-center straight vehicle movement.

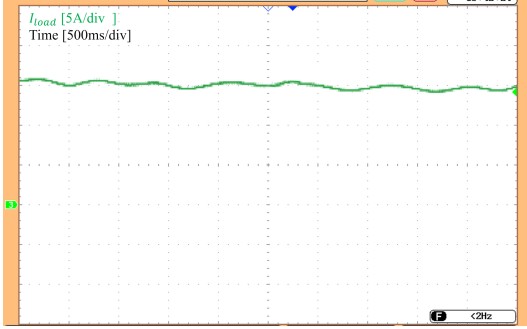

(**a**) Charging current waveform for centered vehicle movement

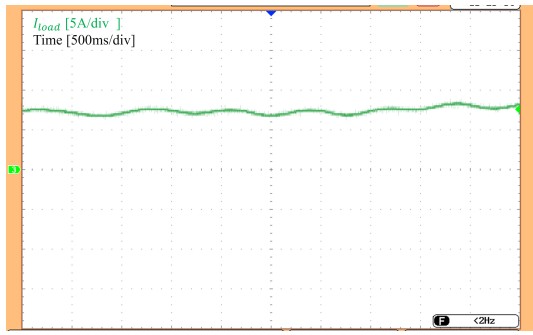

(**b**) Charging current waveform in misaligned vehicle conditions

**Figure 15.** Charging current waveforms for centered and laterally offset vehicle paths.

## 4. Conclusions

This paper proposes a segment-switching method for dynamic wireless charging systems of electric vehicles that eliminates the need for position sensors. Instead, the approach relies on current signals from the compensation inductors, under the ideal condition that the vehicle travels in a straight line while maintaining a constant distance from the charging surface during motion. A modular transmitting coil structure was designed to enable flexible activation and deactivation of individual segments based on the vehicle's position. The coil model was developed using finite element analysis in Ansys Maxwell and further refined through system identification in MATLAB. By analyzing the variations in coupling coefficients during vehicle movement and their impact on the charging current, the study identified optimal switching positions that effectively minimize current pulsations during segment transitions. The equivalent circuit model was also investigated to determine clear indicators for triggering segment activation and deactivation. Current signals from the primary compensation inductors were measured, compared, and used as the control input for switching decisions. The proposed method was validated through both simulation and laboratory experiments. The results demonstrated that the current ripple at the optimal switching points was limited to 2.2% in simulation and 3.3% in experimental measurements, confirming the effectiveness of the approach in maintaining a stable charging current during segment switching. Future work will focus on extending the proposed method to more realistic operating conditions, including misaligned vehicle trajectories and varying coupling distances, as well as implementing real-time control strategies for high-speed vehicle operation.

**Author Contributions:** Formal analysis, T.D.H.; methodology, N.K.T. and N.T.D.; software, N.H.M.; supervision, T.T.M.; validation, T.T.M.; writing—original draft, T.D.H.; Writing—review and editing, N.K.T. and N.T.D. All authors have read and agreed to the published version of the manuscript.

**Funding:** This research was funded by Hanoi University of Science and Technology (HUST) under project number T2023-PC-026.

**Data Availability Statement:** The data generated during the current study is available from the first author or corresponding author on reasonable request.

**Acknowledgments:** The authors thank the reviewers for their helpful and insightful comments.

**Conflicts of Interest:** The authors declare no conflicts of interest.

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
