# Peer review of "Segmentation Control in Dynamic Wireless Charging for Electric Vehicles"

_electronics, doi:10.3390/electronics14153086_

Round 1
Reviewer 1 Report
Comments and Suggestions for Authors
- As shown in Table 2, why is such a small compensation inductance adopted, and with such a small inductance, would a small change in self-inductance at the transmitter result in a large reactive power?
- Some of the units in table 2 are incorrect.
- Due to the soft start time and communication delay of the inverter, is it possible to switch within the range of the given point at high speeds? Or is the application scenario of this thesis a low speed condition?
- Does the switching criterion still hold when a lateral offset occurs and ILft6 > ILft4?
- What is the meaning of GS pulse?
Author Response
We sincerely appreciate the valuable comments and suggestions provided by the reviewers, which have greatly contributed to improving the quality and clarity of the manuscript. We have carefully considered all the feedback and made corresponding revisions, as detailed in the attached response file.

Reviewer 2 Report
Comments and Suggestions for Authors
- The presence of a ferrite sheet acts like a core for the transmitter coils. The ferrite is a nonlinear material, thus the inductor is nonlinear. The principle of superposition can be applied only for linear circuits, namely those containing only linear circuit elements.
- Where is the origin of the x-axis? A reference for x must be given.
- What are the other quantities appearing in (3) and (4)? They are not explicitly defined and the reader is supposed to make assumptions over them.
- How do the authors define the "total coupling coefficient"? The term first appears at line 169, without any indication or definition.
- The authors wrote: "The coupling coefficient function between the two transmitting coils is omitted due to its position-independent behavior and small magnitude." No it isn't omitted. It is given by (7), just before the quoted phrase.
- If the quantities defined by (9) and (10) are the actual "total coupling coefficients", it has to proven their appearance in (8), which in turn must be demonstrated.
- There is no proof that Δk(x) has the two formulas given by (11) and (12). The generally accepted definition for the coupling coefficient is a value between 0 and 1. In (12), to be true, either all three terms are simultaneously 0, or there is at least one negative term. Since Δk(x) is positive, being given by (11), at least one of the other two must be negative, fact that is impossible.
- Eq. (13) doesn't provide a current as mentioned in the preceding phrase. With (11) and (12), one will get 1/2 of the presented right-hand side expression.
- Eqs. (14) and (15) are not proven or commented.
- If Eq. (16) is true, if there is no current extracted form the Uinf sources, according to Kirchhoff's current law.
- The phrase "Consider a point on the charging lane where the vehicle is located and at that location there are n transmitting coils (from the i-th transmitting coil to the (i+n−1)-th transmitting coil) turned on corresponding to the vehicle."..."turned on" by what "corresponding to the vehicle"? The phrase is incomplete.
- "Where, Req1 , Req2 are denote [...]." Either "are" or "denote".
- The symbol of kilometer is "km" not "Km".
Author Response

(The authors gave the same response as above.)

Reviewer 3 Report
Comments and Suggestions for Authors
This paper proposes a position-sensorless control method for managing transmitting lines in a dynamic wireless charging system.
The topic of the paper is highly interesting. Some typical issues should be addressed:
- Some typos are evident for example show to shown in line 104
- Figures should be positioned just after the actual in -text reference.
- Where are the equations from? Is there a reference from another paper?
- Parameters at equations 5-7 should be explained. This explanation can be implemented in table 3
- What are the best coupling positions at figures 3-4 ? any comments ?
- What is the Δκ value, no explanation is evident for a typical reader
- Figure 14 is very explanatory and should be promoted to key factor
- Figure 15 illustrated that 1300mm is the ideal distance , what about efficiency in %?
- Change conclusion to Conclusions
- Conclusions seem relatively small , should be enhanced
- Please revise contribution with only first letters of your names ; eg N.H.M
- Recheck references format style with Zotero , doi is missing and the format for example to paper 24 should be : ‘’Wang, H.; Cheng, K.W.E. An Improved and Integrated Design of Segmented Dynamic Wireless Power Transfer for Electric Vehicles. Energies 2021, 14, 1975. https://doi.org/10.3390/en14071975’’
Very good paper , more clarification and explanation should be provided to enhance readability
Author Response

(The authors gave the same response as above.)

Round 2
Reviewer 1 Report
Comments and Suggestions for Authors
I have no more comments.
Author Response
We would like to express our sincere gratitude to the reviewer for their constructive feedback and thoughtful comments. As the reviewer has no further concerns, we hope that the revised manuscript meets the expectations and is now suitable for publication. Thank you again for your kind support throughout the review process.
Reviewer 2 Report
Comments and Suggestions for Authors
The authors properly addressed my comments by taking the appropriate measures for improving the revised manuscript. There are minimal actions needed for the final manuscript acceptance for publication. For the sake of clarity, it is important to specify if the quantity M from (1) may take negative values.
Author Response
We are grateful to Reviewer 2 for their positive assessment of the revised manuscript and for the additional remark concerning the mutual inductance between the coils. We have carefully considered this suggestion, provided a detailed response in the attached file, and revised the manuscript accordingly.
